# IR-UWB Sensor Based Fall Detection Method Using CNN Algorithm

**DOI:** 10.3390/s20205948

**Published:** 2020-10-21

**Authors:** Taekjin Han, Wonho Kang, Gyunghyun Choi

**Affiliations:** Graduate School of Technology & Innovation Management, Hanyang University, Wangsimni-ro 222, Seongdong-gu, Seoul 04763, Korea; tjhan07@hanyang.ac.kr (T.H.); whkang@danusys.com (W.K.)

**Keywords:** IR-UWB radar sensor, fall detection, fall/ADL classification, deep learning classifier, convolutional neural network

## Abstract

Falls are the leading cause of fatal injuries in the elderly such as fractures, and secondary damage from falls can lead to death. As such, fall detection is a crucial topic. However, due to the trade-off relationship between privacy preservation, user convenience, and fall detection performance, it is generally difficult to develop a fall detection system that simultaneously satisfies all conditions. The main goal of this study is to build a practical fall detection framework that can effectively classify the various behavior types into “Fall” and “Activities of daily living (ADL)” while securing privacy preservation and user convenience. For this purpose, signal data containing the motion information of objects was collected using a non-contact, unobtrusive, and non-restraint impulse-radio ultra wideband (IR-UWB) radar. These data were then applied to a convolutional neural network (CNN) algorithm to create an object behavior type classifier that can classify the behavior types of objects into “Fall” and “ADL.” The data were collected by actually performing various activities of daily living, including falling. The performance of the classifier yielded satisfactory results. By combining an IR-UWB and CNN algorithm, this study demonstrates the feasibility of building a practical fall detection system that exceeds a certain level of detection accuracy while also ensuring privacy preservation and user convenience.

## 1. Introduction

Falls are the leading cause of fatal injuries in the elderly, such as fractures. A fall can lead to a loss of consciousness due to the impact, or conversely, a loss of consciousness may lead to a fall, thereby resulting in death in severe cases. As such, the risk of falls in today’s aging society is a crucial issue. According to the World Health Organization [1], between 28% and 35% of the population aged 65 years and older experience at least one fall each year. These falls account for at least 50% of the causes of hospitalization of the elderly, and approximately 40% of their non-natural causes of death. Furthermore, medical analyses of the damage caused by falls have demonstrated that this is highly dependent on the response and rescue time [2]. That is, the earlier the occurrence of the fall is discovered, the lower the chance of fatality due to secondary damage. Therefore, early detection of falls in the elderly who live alone in isolated environments plays a vital role in preventing the phenomenon of “lonely death” in the elderly. Lonely death refers to the phenomenon of a person dying alone and remaining undiscovered for several days, in which the person typically lives in a house and has no one to nurse or care for him or her.

In the academic field, the detection of falls—a fatal risk to the elderly—has attracted the attention of numerous researchers over the past two decades, representing one of the most important research topics in daily health-care for the elderly [3]. Though fall detection research shows promise, it faces many challenges. The three most important considerations in fall detection studies can be summarized as follows.
High performance of detection algorithm: accuracy, sensitivity, specificity, F1 score, etc.User convenience.Privacy preservation.

The main goal of fall detection systems is to distinguish between fall events and activities of daily living (ADL) [4]. As certain ADLs, such as moving from a sitting or standing posture to a lying posture, are similar to falls, it is difficult to develop an algorithm that accurately distinguishes between them. In addition to the high performance of the fall detection algorithm, the most important issues to address are user convenience and privacy preservation. User convenience refers to freeing the user of physical constraints by eliminating the need to wear measurement sensors on parts of the body, and to performing unobtrusive measurements at any point in time unbeknownst to the user. Privacy preservation is a crucial factor due to ethical issues that may arise from infringement on the privacy of the individual in their living space, such as in their bedroom or bathroom.

To solve problems related to the above considerations, as shown in Figure 1, numerous researchers have introduced a variety of sensors to measure falls or have changed the type of algorithm used to judge events. Changing the sensor type directly impacts all or some of the main issues, whereas changing the algorithm used to judge events compensates for the limitations arising from low performance when using the same sensor. Researchers have thus far conducted the following studies and approaches on sensors and detection algorithms for fall detection.

### 1.1. Sensors

Beginning with fall detection using accelerometers [5] up to the recent skeleton-based measurement using Microsoft Kinect sensors [6], methods utilizing various sensors for fall detection have been attempted. Since the start of fall research using accelerometers in 1991 [5], wearable devices were the most widely used sensors in early studies. This method involved the principle of measuring the variation in the rapidly changing acceleration of an object when a fall occurs. According to Igual et al. [4], among 197 studies using wearable devices to detect falls, 186 used accelerometers. These studies measured falls through various methods, including using only a three-axis accelerometer [7], increasing the number of accelerometers to compensate for accuracy [8], or adding sensors such as a barometer [9] or gyroscope [10,11]. Fall detection methods using wearable devices have the advantages of cost efficiency, privacy preservation, and ease of sensor installation and data collection. However, these methods degrade user convenience, as the user must wear or attach sensors to a part of the body such as the waist, head, chest, thigh, knee, underarms, ear lobe, or wrist. Wearable devices are able to measure more activity indicators in addition to falls (e.g., activity, heart rate) with higher accuracy. However, due to the nature of the elderly—the primary users—it is difficult to wear these devices continuously, and their battery life is also limited. Moreover, they must be charged by a third party such as a social worker or guardian on site. In addition, due to the stigma against weakness in the elderly, industrial development has not progressed [12,13,14]. Meanwhile, to address the disadvantages in terms of user convenience in wearable devices, researchers have investigated the use of smartphones embedded with an accelerometer as sensors. According to Ren and Peng [15], smartphones always have built-in sensors such as three-axis accelerometers, gyroscopes and magnetic, high microprocessors, etc., which makes smartphones a very good platform to detect a human fall. Though research in this area has been steadily growing since the first study in 2009 using smartphones [16], given that the primary users are the elderly who are still unfamiliar with smartphones, industrial development has been difficult. Moreover, the smartphone battery life and battery consumption may impede usage. (Casilari-Perez and Garcia-Lagos [17] also stated that there are several drawbacks given that fall detection solution must coexist with the complex heterogeneous application running concurrently on a smartphone.

Fall detection methods using camera- (vision)-based sensors are some of the most widely applied methods, Espinosa et al. [18] used a background subtraction technique to separate the object (person) in the image from the background, generated a bounding box by connecting the furthest foreground pixels, and extracted the aspect ratio of the person. Through this, the normal posture (standing up straight posture) and abnormal posture (fall) were distinguished. Rougier et al. [19] used an ellipse rather than a bounding box in special situations such as falls or the person’s daily activities, and the direction standard deviation and ratio standard deviation of the ellipse were used to judge the event of fall. Meanwhile, Chua et al. [20] used only three points to represent the person rather than a bounding box or ellipse. This fall detection technique involves using the extracted features to analyze changes in the shape of the person’s silhouette through information on changes in the upper and lower parts of the human body. Camera-based fall detection methods exhibit higher detection accuracy and robustness than other technologies [18]. Moreover, they cause less disturbance to daily life than wearable sensors. However, as mentioned in numerous studies such as Igual et al. [4], the fatal weakness of video-based fall detection solutions is their invasion of privacy. Continuously surveilling private spaces such as the bedroom and bathroom through camera- (vision)-based systems can result in ethical and legal issues related to privacy infringement and the protection of personal information. Additionally, camera-based measurement methods suffer from degraded performance in dark environments, unless a special-purpose camera (e.g., infrared) is used. According to Xu et al. [3], since 2014, the number of studies related to fall detection systems using cameras has greatly decreased, and studies using 3D depth cameras, represented by Microsoft Kinect, have rapidly increased to address the privacy-related disadvantages of camera- (vision)-based sensors. However, the effective sensing distance of the Kinect sensor is between 0.4 m and 3 m, introducing physical limitations for use in real life.

There are also ambient devices that detect falls, in which at least one sensor is placed in the user’s living space to collect information on the user’s interaction with the sensor(s). Ambient devices include pressure sensors, radio frequency (RF)-based sensors, and floor vibration-based fall detectors, and have the following features. Pressure sensors, which are often used for their low cost and non-invasiveness, use the basic principle that pressure increases as the user approaches the sensor. RF-based sensors, which include Doppler and IR-UWB sensors, use the information from objects contained in the changes in frequency, phase, and arrival time of signals received by the movement of the object to detect respiratory rate, heart rate, movement, etc. Floor vibration-based fall detectors use the vibration or pressure caused by a person’s body colliding with the floor to detect fall events [21]. Acoustic sensors attached to the floor capture the sound generated from falls to detect falls [22]. As the sensors are installed only inside the home environment, they present no inconvenience to the user such as attachment to the body, thus allowing the user to naturally and continuously perform daily activities [23]. Despite the advantages of privacy preservation and user convenience, however, these methods generate many false alarms and have trouble detecting falls outside the installed area (see Table 1).

Despite the advantages of the methods described above that use wearable devices and camera-based sensors, they have fatal disadvantages in terms of user convenience and privacy. Indeed, Wild et al. [24], found that home sensing technologies that are unobtrusive to the elderly, do not require the user to learn new technologies, and most importantly, do not capture video images, are better accepted. As such, fall detection systems must be constructed based on sensing technology that satisfies the conditions of unobtrusiveness, lack of need to wear the device, and privacy preservation, which are requirements for the elderly to accept the technology.

### 1.2. Algorithms

Meanwhile, studies for enhancing the performance of fall detection systems also involve changes to improve the performance of the event detection algorithm according to sensor development, in addition to various attempts at the sensor level [3]. Early wearable device-based fall detection studies primarily used threshold-based detection algorithms. Many studies before 2010 used techniques based on threshold value for automatic fall detection [25,26]. Threshold-based detection algorithms are relatively easy to apply and implement due to their simple operation principle; however, when expertise is insufficient, it is difficult to derive a threshold suitable for specific situations. According to Mrozek et al. [27], the result of threshold-based methods is prone to high variability for various groups of monitored people. Due to the difficulty of ensuring the robustness of the algorithm when applying the same reference value in various environments and situations, each environment and situation must be configured separately to optimize the detection algorithm’s performance.

However, the detection technologies used in detection algorithms vary with sensor type. With new types of advanced sensors being increasingly introduced, researchers are further introducing machine learning-based algorithms to more accurately detect human activity in diverse environments. According to Ren and Peng [15], from an analytical algorithm perspective, the threshold-based method is a classical and basic approach that compares with a reference value, while machine learning-based method has been widely researched to increase the accuracy of the system. Machine learning involves analyzing given data to find regularity, patterns, etc., and using this to extract meaningful information. Traditional machine learning algorithms widely used for fall detection include support vector machine (SVM) [28,29,30], decision tree (DT) [30,31], scale invariant feature transform (SIFT) [32], and histogram of gradient (HOG) [33]. However, these techniques require an expert to devote attention to data preprocessing and parameter setting to extract appropriate features of the data for transmission as input in the machine learning algorithm. Their most fatal weakness is that if the features extracted by the person are not suitable for recognition, it is difficult to achieve good recognition performance no matter the machine learning algorithm used.

Various deep learning algorithms to address this disadvantage have recently emerged. Recently, deep learning has shown incredible capabilities in areas such as computer vision, video processing, and natural language processing [34]. Many researchers have achieved good results by adopting deep learning algorithms in fall detection studies. In deep learning algorithms, a person does not directly extract features such as patterns or rules, as in traditional machine learning algorithms. Rather, the algorithm uses feature representation learning, which enables computers to capture features on their own, to extract and utilize features that a person cannot recognize. Accordingly, deep learning algorithms exhibit a higher performance than existing algorithms. In particular, convolutional neural networks (CNN) can learn the unique features of objects regardless of changes in image position, size, and angle, thus making CNN invariant to movement, noise, and distortion in the image and enabling classification even when the size or shape changes.

### 1.3. Characteristics of IR-UWB Radar Sensor and Related Works

IR-UWB radar, a type of RF sensor, uses the time-of-arrival to measure the distance to an object. The time-of-arrival is the travel time of an extremely short impulse signal with an ultra wideband (UWB) radiated from a transmitting antenna to hit the target and reflect back to the receiving antenna. Using the pattern of changes in signals affected by the movement of objects, researchers have studied the positioning of objects [35,36], human detection [37], measurement of bio-signals such as respiration and heartbeat rates [38], identification of non-line-of-sight condition [39], mobile phone usage detection [40], hand gesture inference [41], and hand gesture recognition based on finger-counting [42].

Moreover, IR-UWB radars have several technical advantages such as accuracy during indoor positioning, robustness to temperature and humidity, and measurement in invisible situations, making them highly suitable for measuring falls in the elderly in their living space. Object behavior type recognition using IR-UWB radars has no issues related to privacy preservation and also secures user convenience because it is a non-contact method.

Due to the advantages of the IR-UWB radar sensor, the number of studies for detecting falls using IR-UWB in recent years is on the rise. In a relatively early study [43], a study that detects various movements including “fall” was performed, and a detection rate of 95% was obtained. The test subject had stayed in the same place for a long time after he fell down, and it was estimated that a “fall” occurred by measuring the physical distance with the IR-UWB. As inference of movement type was made based on the distance between the test subject and sensor according to scenario, it was difficult to draw inference when there was a change in movement over distance and space. In the study of Mercuri et al. [44], high accuracy, precision and sensitivity were obtained, but there was a disadvantage in that the experiment and detection were conducted only for frontal falls directed to the IR-UWB. In the study of Mokhtari, et al. [45], a machine learning algorithm based on unsupervised learning was applied experimentally, but there is a disadvantage in that the IR-UWB sensor was mounted over the door frame and movement activities were detected only in the area under the door. In the study of Tsuchiyama et al. [46], an IR-UWB sensor was attached to the back side of toilet lid. By perceiving the fact that the signal was blocked or weakened when there was a human on the toilet seat or outside the door, the author detected the type of activities effectively. However, the measurement area was too small (1 m × 1.4 m) and limited. Moreover, it could be difficult to achieve accuracy of 95% when “Fall” occurred in a large space without objects. As a similar study, there is also a study that detects “fall” using Wi-Fi, an RF-based sensor. Wang et al. [47], used only commercial Wi-Fi product to detect motion, so the equipment cost was very low. WiFall achieved an average fall detection accuracy of 94% using the random forest algorithm in all test scenarios. However, the false alarm rate tends to be rather high, with an average of 13%.

### 1.4. Our Approaches

The main goal of this study is to build a fall detection framework that can effectively classify the behavior of an object into “Fall” and “ADL” through a deep-learning based algorithm while using IR-UWB sensor for data collection to ensure privacy preservation and user convenience.

The radar sensor-based fall detection method is based on the development of a model to extract a series of features from signal data, and to distinguish falls from daily activities. In general, to extract the characteristics of radar signal data, a mathematically complex preprocess, which requires expertise in the time-frequency domain, is required. This is not different even in using traditional machine learning-based systems, researchers must directly program features to be recognized. The features to be extracted are limited in type, and researchers wishing to perform manual engineering are required to have practical experience and abundant expertise.

Therefore, this study explores a method of applying IR-UWB radar sensor data to CNN, one of the deep learning algorithms, to improve performance problems that depend on researchers’ manual characterization results. CNN shows excellent performance when the input data include specific patterns or changes in patterns. A CNN-based fall detection system can detect fall events by converting fall and ADL-related sensor data into appropriate images that can be classified with CNN. When performing daily activities such as falls, one-dimensional signal data changes due to the instantaneous change of distance between IR-UWB radar sensor and a person. It is required to convert the pattern of the signal data changes into a time-distance two-dimensional image representation that can effectively contain the characteristic pattern. After converting IR-UWB signal data into an image that is a suitable form for applying it to CNN algorithms, using the power of CNN, which shows strength in image classification, the occurrence of a fall can be detected by classifying the image generated with the change pattern of the radar signal caused by the behavior of the object.

To verify the effectiveness of the development framework, the IR-UWB and internet protocol (IP) camera are used to collect the motion data of objects that perform various types of behaviors, including fall at different distances. Then, to apply the collected IR-UWB signal data to the CNN algorithm rather than the conventional complex signal processing process, the data are preprocessed into a suitable form. Through automatic feature extraction and learning of signal changes due to the object’s motion using the CNN algorithm, a classifier for the behavior types of the object is generated. Data not used for learning are then input into the classifier to verify its performance in classifying the behavior types (fall/ADL) of the object, thereby demonstrating the effectiveness of the developmental framework. To collect motion information of the training and test-related objects, six volunteers participated in the experiment of this study, taking gender into account.

In summary, the main contributions of this work are as follows:Proposed a preprocessing method that converts one-dimensional time series signal data of IR-UWB radar sensor to a distance-time two-dimensional image, which contains features of signal change effectively.
-Decision of standard frame of pixel figure difference, decision of motion image window size, proposition of labeling method.Proposed to use generated 2D grey scale image by CNN algorithm for fall detection.
-Extraction and learning of signal change pattern of radar sensor by CNN algorithm automatically.Proposed a decision algorithm that can decide the type of event by reviewing the classification result of a single image.
-Replaced problem of image classification for problem of fall detection and proposed the image judgement ratio that shows peak performance of an experiment.

This paper is structured as follows. Section 2 proposes a framework for developing a simple fall detection algorithm based on an IR-UWB sensor and CNN algorithm. This section describes the data collection and preprocessing methods, establishment of the machine learning model, and a fall detection method based on the CNN algorithm image classification method. Section 3 describes the test environment configuration to verify the proposed technique, and Section 4 introduces and discusses the performance evaluation results of the developed fall detection algorithm. Finally, Section 5 concludes the study and discusses possible future research directions.

## 2. Materials and Methods

### 2.1. System Overview

The framework proposed in this study comprises three main steps as shown in Figure 2; motion and data acquisition, data preprocessing, and CNN-based feature extraction and classification. In detail, the framework consists of three stages: acquiring object behavior data using the IR-UWB and IP camera (stage 1); preprocessing the acquired data into a form suitable to apply to the CNN algorithm (stage 2); automatic feature extraction and learning for signal changes of the IR-UWB when a fall event occurs using the CNN algorithm, and using the trained CNN model to finally judge whether “fall” occurs through image recognition and classification (stage 3).

Without a conventional complex context-dependent signal processing step for each target (action or event) to be detected, the IR-UWB data are imaged through preprocessing and applied to the CNN algorithm, and the classification performance for object behavior type is verified through the generated CNN-based object behavior type classifier. Through this, this study demonstrates the effectiveness of the development framework.

### 2.2. Stage 1: Motion and Data Acquisitions

The five motion types of the objects to be measured by the IR-UWB radar sensor in this study include “walk”, “sit”, “stand”, and “rise”, which are typical indoor activities of daily life of the elderly, as well as “fall”, which is an abnormal behavior type. Table 2 shows the class classification and motion-related detailed definition for each behavior type to construct an object behavior type classifier learning database that can distinguish between “Fall” and “ADL”. To develop a detection algorithm that exhibits a more robust performance when performing experiments to construct the database, the scenarios included performing various types of behaviors at different distances from the sensor, as well as walking in a variety of directions. Volunteers aged younger than 25 participated in the actual experiment, and the samples were constructed considering the gender ratio. To develop an algorithm that shows more a robust performance in diverse situations, the participants were guided in advance to freely execute a “sitting” direction, “standing” direction, “falling” direction, and “time required to fall”.

Meanwhile, the movement of the object was measured simultaneously using the IR-UWB radar sensor and IP camera, as shown in Figure 3. In this study, the change (amount) pattern of the signal power value of the IR-UWB signal reflected from the movement of the object was used for fall detection. The IR-UWB sensor collects data at 24 frames per second comprising 660 signal power values per frame. In the scenario, an alarm is sounded to signal the start of a behavior, though the participants were allowed to freely decide the direction or time taken to perform the behavior. To improve the accuracy of data labeling accordingly, in addition to an IR-UWB radar sensor, an IP camera was used to collect additional images, and the time that the event occurred was also used as a supplement during labeling. The IP camera was limitedly used to acquire (supplement) labeling information only when creating the machine learning-based event classifier, and after generating the “Fall/ADL” object behavior type classifier, only the object movement was measured through the IR-UWB without the IP camera.

### 2.3. Stage 2: Data Pre-Processing

To judge whether a fall occurs by applying the CNN algorithm to the IR-UWB sensor data collected through the experiments, a method is required to preprocess the data into a form suitable for machine learning application. IR-UWB signal data should be mapped to a 2D visualization image, which is an appropriate form that can be put as an input of the CNN algorithms through preprocessing. The IR-UWB data preprocessing method proposed in this study comprises four steps:Perform inter-frame subtraction after 24th frame;Determine size of motion image;Outlier replacement and normalizing;Grey scale image visualization and image data labelling for CNN algorithm.

In the first step, using the core principle of the difference in pixel values between frames in the background subtraction technique, which is extensively used to separate the foreground and background in video analysis, the signal features that change due to object movement were extracted. The “fall” behavior of the person indicates that the distance between the person and the IR-UWB radar sensor changed greatly over a short period of time. In this study, the change in IR-UWB signal generated by the object’s movement was extracted through the residual between the current frame and the frame after a certain period of time. This change pattern was then utilized as a feature to detect falls. It is also possible to obtain the effect of canceling the signals reflected by a stationary object by subtracting the signal power values between frames. Meanwhile, the amount of change in the signals rather than the signals themselves were extracted because the signal frame data of the IR-UWB contain only information that can estimate the position of an object or whether it is moving at that point in time. Hence, it is difficult to judge the behavior type of the object.

Here, it is important to decide on which subject frame should be subtracted from current frame. In other words, an interval must be set to maximize the detection effect of the change in signal generated by the fall motion as a result of subtraction. Therefore, the case in which shows a high figure of amplitude of signal change at the point of fall and lower figure of amplitude of signal change than the threshold value at the points unrelated to the occurrence of fall was finally selected. The amount of change in the signal, which is the result of subtraction between the frames, can be expressed as h(x) in Equation (1). In Equation (1), the experiment was performed by changing n to six stages: 1 frame (1/24 s), 12 frames (0.5 s), 24 frames (1 s), 48 frames (2 s), 72 frames (3 s), and 96 frames (4 s). As in Equations (1) and (2), when the change in the signal is greater than the threshold (C), the change is set to h(x), and if smaller than the threshold, then it is set to 0. When a fall occurs, the amplitude of change in signal greater than a certain value is expressed at the location the fall occurred. Figure 4 shows a visualization of the change in signal obtained by calculating the residual value between the current frame and the frame after the 24th frame according to Equations (1) and (2) when a fall occurs.
(1)$$h\left(\chi \right)=Signal\;Power\;Value_{present}-Signal\;Power\;Value_{after\;n^{th}frame}\;where\;n\;is\;1\;to\;96$$
(2)$$h\left(\chi \right)=\{\begin{array}{c} \;10*\left|h\left(\chi \right)\right|,\;\;if\;\left|h\left(\chi \right)\right|\geq C\\ \;\;\;\;\;\;\;\;\;\left|h\left(\chi \right)\right|,\;\;if\;\left|h\left(\chi \right)\right|<C\;\;\; \end{array}where\;C\;is\;threshold\;value$$

An experiment was conducted in which *n* was changed to six stages. According to the results, when using the residual between the current frame and frame after the 24th frame, the amplitude of signal change at the location where fall occurred was large, and the signal amplitude at locations unrelated to the fall was small. Hence, the residual between the frame after the 24th frame was used.

In the second stage, the size of the motion image is determined. A motion image is a two-dimensional image of space-time flow that contains the motion information of an object. Regardless of the space where an action occurs, the intensity of a signal reflected by a wall or object that is too far away from the IR-UWB sensor is weak, thus degrading the performance of the detection algorithm or resulting in unnecessary computation for data analysis. Considering the living space size of elderly persons living alone and the experimental scenario, the size of the measurement space was set to a maximum of the 300th frame (approximately 6 m) from the sensor. Meanwhile, due to the nature of IR-UWB radar sensors, they generate considerable noise values regardless of the object’s movement in samples close to the radar. To remove the portion of data up to the samples with noise values, a cumulative scalar quantity distribution analysis was conducted, as shown in Figure 5, using the absolute value of the signal amplitude for the 660 samples per single frame. Through an analysis of 10 data files, based on the maximum signal value in the samples after the 50th sample, the number of the last sample exceeding the reference value among the samples before 50th sample was obtained. The analysis indicated that considerable noise values were generated up to the 44th sample regardless of the object based on the average distance from the IR-UWB sensor. Accordingly, the final data used were that between the 45th and 300th samples.

It is necessary to determine the minimum length that must be included to sufficiently reflect the object’s behavior type information related to the vertical length of the motion image. For this purpose, this study performed an experiment to repeatedly measure the average time taken for a person to sit or fall on the floor from a standing position. According to the measurements, the average time taken to sit or fall on the floor was approximately 4 s, hence it was decided to include at least 96 frames (at least 4 s). Furthermore, the experimental scenario was configured to give free time of approximately 3 s before and after certain behaviors such as “sit” and “fall”, thereby allowing separation between certain behaviors. Giving free time involves placing certain behaviors before and after the occurrence of a fall such as “stand” and “lie down into a fallen state.” Considering the above, the motion image size to use as input for the CNN algorithm was finally set to 256 × 256.

The third stage is to normalize and replace outliers in the signal change data. As outliers are excessively outside a certain range and greatly impact the results of data analysis or modeling, they must be replaced. Due to the nature of RF sensors, the amplitude of the IR-UWB radar signal varies over a very wide range and outliers inevitably occur. Before performing gray scale image visualization, the outliers are replaced and normalization is performed to limit the data to a uniform range. Isolation Forest in the sklearn package of Python was used to detect the outliers. Then, after determining the lower and upper limits, detected outliers below the lower limit were replaced with the lower limit value, and outliers above the upper limit were replaced with the upper limit value. Then, 3 sigma was applied to replace outliers corresponding to more than 99.7% and less than 0.3%. After outlier replacement, to recognize the form of change in IR-UWB signal, which varies over a wide range, the data must be normalized within a certain range based on the maximum and minimum values of the amount of change in signal. This study normalized the data between 0 (minimum) and 255 (maximum) according to the intensity, based on which grey scale images were generated in which, pixels at minimum intensity are black, pixels at maximum intensity are white, and pixels in between are shades of grey. When establishing the fall/ADL object behavior type classifier model, grey scale motion images with 256 intensity values generated based on the change in IR-UWB signals were used as input for machine learning. Signal power value data were collected when the participant walked towards and away from the IR-UWB sensor in a round trip, to apply these data to the CNN algorithm, it was processed into a meaningful form and imaged as shown in Figure 6. As shown, the pattern of changes in signal varies as the object approaches and moves away from the IR-UWB sensor.

In the last stage, to prepare a supervised learning dataset to create the fall/ADL object behavior type classifier, the behavior type information was labeled for each generated grey scale motion image. During data collection, while viewing the additional images captured by the IP camera to construct a machine learning database, the behavior type of each object (walk, sit, rise, stand, fall) was labeled in each image according to each event definition for the time at which the event occurred and the event type. Based on the information of event occurrence time in the camera images, the behavior types signify the following based on the IR-UWB signals and each event: “fall” indicates “time at which the head touches the floor after falling”, “rise” indicates “time at which the head rises from the floor from a fallen state”, “walk” indicates “time at which walking begins after standing”, and “sit” indicates “time at which the hips touch the floor”. For example, as demonstrated by the repeated measurements, the average time taken for a person to fall is approximately 4 s, hence based on the event occurrence time, the images containing an interval up to 4 s before a fall actually occurred were labeled as “fall”. As shown in Figure 7, since image 1 to image 4 contain the part in which “fall” occurs, these were labeled as “fall”. In Figure 7, 1 frame per second is assumed for the purpose of explanation; in reality, however, 24 frames are generated per second.

### 2.4. Stage 3: CNN Algorithm Based Feature Extraction and Classification

#### 2.4.1. CNN Architecture for Training and Classification

The final stage of the framework proposed in this study involves applying the CNN algorithm to the preprocessed visualization image data to automatically extract and learn the characteristics of the changes in IR-UWB signal that arise when a fall event occurs. The trained pattern is then used to finally judge whether a fall occurred through image recognition and classification. For this purpose, the dataset consisting of motion images and labeling information generated through data preprocessing is put into a trainer to perform machine learning to create the fall/ADL object behavior type classifier. As the objective is to generate a CNN-based object behavior type classifier (fall/ADL) that can detect fall, “fall” was considered one class in object detection (fall), while stand, rise, etc., were considered as a different class (ADL).

The structure of the CNN network implemented for characterization and classification of visualized motion images in this study is as shown in Figure 8. The structure of CNN was based on LeNet-5 [48]. We constructed the CNN network to extract image features through two convolution layers and two pooling layers, and to classify images in fully connected layers.

The size of the input image of the CNN network is 256 × 256. The main purpose of CNN’s first step, the convolution phase, is to extract features from input images. In the convolution layer, a 5 × 5 convolution filter was used, and the stride was 1. The feature map generated as a result of the convolution is smaller than the original image, but has the advantage of being able to process easily and quickly while detecting the features of an image. After each convolution layer, batch normalization is performed to reduce overfitting and speed up the training process. Additionally, to give the model nonlinearity, a rectified linear unit (ReLU) layer was applied as an activation function for the convolutional feature at the end of each operation. This is because CNN’s input images have nonlinear characteristics. ReLU is known to be able to train networks faster than other operations, such as tanh or sigmoid, without significantly affecting generalization accuracy. For subsampling, 2 × 2 max pooling was used, and the strides were 2. By taking the maximum value at each part, it removes 75% of the information that is not characteristic and helps to reduce the number of required parameters and calculation quantities. In addition, the model built through the CNN can be controlled from overfitting by max pooling. The main purpose of the fully connected layer is to combine the extracted features into more properties so that classes can be predicted with higher accuracy. The softmax activation function outputs probability values between 0 and 1 for the classification labels that the model wants to predict. Since the machine learning model developed in this study is a binary classifier, the fully connected layer outputs a two-dimensional vector, which contains the probability for each class of images to be classified. The output indicates which label the image is most likely to have.

#### 2.4.2. Performance Evaluation Levels: Single Image and Single Event

The final goal of this study is to ensure that the trained classification models correctly classify fall and ADL events. After the training of the model is completed, the established model can be evaluated by classification results with unused images (test dataset). The performance of the model can be evaluated by how accurately the trained algorithm can classify “fall” and “ADL” by checking the classification results for two evaluation levels: (1) single image classification and (2) single event classification. First, for single image classification, each image is classified as “fall” or “ADL” by using the CNN-based model as shown in Figure 9a, and the model is evaluated based on the classification result. On the other hand, in the case of single event classification, it is carried out according to the procedure as follows. The images generated consecutively by one action within the same experiment are grouped together. One group consists of at least 60 images, and if there are less than 60 images the event group is not created. A created single event group corresponds to one actual behavior in the experiment performed by the test subject. If images belonging to the same group are classified to be a “specific” event by a certain percentage or more, the event group is finally determined to belong to the “specific” event. Performance is evaluated by comparing classification results with ground-truth.

#### 2.4.3. Performance Evaluation Indicators

Suitable performance evaluation indicators must be selected to properly assess the performance of the generated fall/ADL object behavior type classifier. If this CNN-based binary classifier properly recognizes the fall, then fall detection becomes positive, and if the classifier does not recognize the fall, then it becomes negative. Since the output is binary, the quality of the fall/ADL object behavior classifier cannot be evaluated through a single test; rather, statistical analysis through a series of tests is needed. To assess the performance of the recognition/detection technology, the recall and precision must be simultaneously considered in addition to the accuracy, which indicates how close the system’s result is to the true value. Recall indicates how well the target of detection (e.g., object or event) is captured without being missed, and precision indicates the accuracy of the system’s detection results, i.e., how many actual falls are included in the detection results. In particular, given that a fall can lead to death, there is a high cost associated with false negatives (that is, the system predicts a fall to have not occurred when a fall actually did occur). Therefore, the figures related to recall must be considered. 

The recall and precision of algorithms generally share a trade-off relationship. As the recall increases by adjusting algorithm parameters, the rate of false detection increases, and by strengthening the conditions to reduce false detections, recall also decreases. Therefore, to properly compare and assess the performance of the recognition algorithm, as shown in Table 3, changes in performance of precision and recall must be taken into account. Accordingly, the F1 score was used as the final performance evaluation indicator, which considers both recall and precision. The F1 score is calculated by the harmonic average of recall and precision. The F1 score is a preferable measurement technique when the balance between precision and recall must be found, and particularly when there is a non-uniform class distribution. Accuracy is not always the best measure for selecting a model. Since the accuracy is not a robust measure in the case of unbalanced data (generally the case of fall detection), the F-score, which is more credible is such a case, was also retained to ensure a fair performance investigation [49].

Same performance indicators are applied for both experiments—single image classification and single event classification.

## 3. Experimental Setup

The experiment was conducted in a laboratory environment. Figure 10a shows the environmental settings. Considering the living space for one person and coverage of IR-UWB antenna, the experimental area was set as 4.5 m × 4.5 m though the full size of the subject area was 7.2 m × 7.2 m. Test subjects conducted activities such as sitting, falling and standing to alarm sound at the “Action Point” shown as “X” in Figure 10a. To prevent accidents during the test, a mat was placed at the point of “Fall”. Figure 10b shows the location of test supervisor. All tests were conducted under test supervisor’s command who sat in front of the door. As the test supervisor was located out of IR-UWB antenna’s coverage, the test supervisor was in place during the test. All the data including video and signal were collected by IR-UWB sensor and IP camera as shown in Figure 10c. The radar sensor is connected to a laptop computer to record the IR-UWB raw data.

To build a database for training the object behavior type classifier to distinguish between “Fall” and “ADL”, processes such as performing various types of behaviors at different distances and walking in various directions were included in the scenarios. The participants were made to repeatedly perform the same type of behavior at specific times according to a recorded notification sound, according to each scenario.

To conduct a steady experiment, test subjects repeated the same kinds of activities to the sound of a recorded alarm at specific time as shown in Figure 11. As per the scenario, an alarm was sounded to the test subjects at the timing of commencement of activities. However, to develop an algorithm that shows robust stability, test subjects were given some leeway in respect of direction as to their activities.

The IR-UWB radar signals and image data captured by the IP camera (labeled information; type of event, event occurrence time) were simultaneously collected and used to construct a machine learning database. Table 4 shows the information of six volunteers participated in the experiment at an average of one hour per person (50% male/female, aged 20–24). About 12 scenarios were conducted per hour, resulting in a basic dataset of approximately 420 scenarios. Figure 12a–d shows the video data collected by the IP camera.

Figure 12e shows the IR-UWB radar sensor used in the experiment. The IR-UWB radar used in the experiment is an ultra-small UWB high-resolution radar sensor that transmits and receives UWB impulses with the HST-S1M-SEmodule of U-main Inc. The HST-S1M-SE module records spatial information (samples) in units of 2.03 cm at approximately 24 frames per second, and can store 660 samples collected in a space up to a distance of 13.4 m. Table 5 shows the detailed specifications of the HST-S1M-SE module.

The CNN code was written using Python syntax through TensorFlow, a deep learning library by Google, and the experiment was conducted on a GTX1080 8G, GPU version. To prevent overfitting, 80% of the data were used as training data, while the remaining 20% not used to generate the training weight were used for testing. Then, 30,000 iterations were performed with an epoch size of 100 and batch size of 50, and in the fully connected layer, 10% of the nodes dropped out. A 5 × 5 filter was used for the convolution layers as the hyper parameters (see Table 6).

This study performed experiments using the test dataset to assess the performance of the fall/ADL object behavior type classifier model generated through machine learning. Accuracy, recall, precision, and F1 score were used as the final performance evaluation indicators (see Table 3).

## 4. Results and Discussion

### 4.1. Motion Image Visualization

The motion information of the objects was collected by the IR-UWB radar sensor and was visualized as two-dimensional 256 × 256 grey scale motion images according to the procedures as shown in Section 2.3. Figure 13 shows the result of visualization, with images corresponding to the five types of activities. The types of motions performed by the participants with different physical conditions (gender, height, weight, etc.) varied slightly in direction and the time taken to perform them; however, under the same event occurrence conditions (position, type), they showed similar patterns. Moving from the left to right, Figure 13a–e represent fall, walk, rise, sit and stand, respectively. The characteristic pattern in image can be used to distinguish fall from other ADL by using a machine learning-based classifier. Even if the subjects’ physical conditions such as height and weight are different, an image with a similar pattern is created.

### 4.2. Database Configuration for CNN-Based Model Construction and Evaluation

#### 4.2.1. Image-Level Database Configuration

The experiment was repeated about 420 times by six other test subjects and finally 447,360 images were generated as result of preprocessing. Images related to “fall” accounted for about 20% of the total number of images. The generated images were classified into each class (fall/ADL) and used as input to train the “Fall/ADL” object behavior type classifier model and assess the performance of generated model. Table 7 shows the configuration and class classification of motion images generated from IR-UWB signal data. Furthermore, 75% of collected data were used for training purposes, while the remaining 25% were used to evaluate the performance of the established model.

#### 4.2.2. Event-Level Database Configuration

As mentioned in Section 2.4.2, for a single event classification, the first requirement is to create event groups with images successively generated by one action within the same experiment. Each event group consists of at least 60 images, and if there were less than 60 images, no event group was generated. The single event group was generated using images from test dataset in Table 7. Table 8 shows the event group creation result.

### 4.3. Classification Results

#### 4.3.1. Classification Results for Single Image

Table 9 shows the single image classification results of the fall/ADL object behavior type classifier. The performance evaluation indicators of the algorithm were calculated based on Table 9. As shown in Table 10, the single image classification result demonstrates a classification accuracy for the test data of 89.65% on average, recall of 74.12%, precision of 75.13%, and F1 of 74.62%.

#### 4.3.2. Classification Results for Single Event: Evaluating the Final Performance of Model

Table 11 shows the event classification results based on the fall/ADL object behavior type classifier. The performance evaluation indicators of the algorithm were also calculated. As shown in Table 12, the classification results demonstrate a classification accuracy for the event group data of 96.65% on average, recall of 92.20%, precision of 88.14%, and F1 score of 90.12%. Given that all the results of classification of consecutive images in event-group were taken into account, sorted type of activities at the single event classification shows an enhanced performance result in comparison to the single image classification. The reason is as follows: assume that among 40 out of 48 motion images containing “Fall” information were classified as “Fall” and rest were classified as “ADL”. Then, according to the single image classification, the algorithm can only achieve an accuracy of 80%. However, in new decision criteria of the single event classification “If the number of images classified as ‘Fall’ for the same event is more than 75%, then a ‘Fall’ has occurred” when 36 out of 48 motion images are classified as “Fall” then the system is deemed to be successful in detecting the occurrence of a fall. Figure 14 shows changes in the final performance evaluation index (F1 score) according to the fall/ADL judgment ratio of the images constituting the event group.

## 5. Conclusions

This study aimed to build a fall detection system that can ensure both privacy preservation and user convenience. To this end, by combining an IR-UWB radar sensor and a CNN algorithm, this study proposed a development framework for a fall/ADL classifier that can classify the behavior types of objects. To interpret IR-UWB radar sensor data using the CNN algorithm, we proposed a preprocessing method that converts one-dimensional time series signal data of IR-UWB radar sensor to a distance-time two dimensional image that contains features of signal change effectively. A classification model was built using the 2D gray scale image generated through preprocessing as an input to the CNN algorithm. We also proposed a decision algorithm that can decide the type of event by reviewing the classification result of a single image. We demonstrated that the CNN-based algorithm achieved an overall accuracy of 96.35%, recall of 92.20%, precision of 88.14% and F1 score of 90.12%, respectively. The performance evaluation results of the classifier obtained in this study demonstrate its feasibility to detect falls using a practical fall detection system combining an IR-UWB and CNN. Using the trained weight, the CNN algorithm can be utilized to detect falls even when another person falls or the position or motion of the fall differs. Though the classification is binary, the results can be considered encouraging given the diversity of the training data and the simplicity of the applied method. It also has been demonstrated that signal data from IR-UWB radar can be mapped to distance-time 2D gray scale images.

Factors that can lead to a failure in classifying the behavior type include the following. Falls are unusual activities that occur infrequently compared to normal daily activities, it is difficult and dangerous to collect actual data of falls [50]. This causes an imbalance in the amount of image data belonging to the classification class, and the imbalance in the amount of data caused the CNN algorithm to learn more about some behavior types. Moreover, though the CNN structure used in the experiment was built based on LeNet-5, it still may not be suitable for effectively discriminating large amounts of images. Finally, due to the nature of the deep learning algorithm, classification accuracy can be improved by increasing the number of data in the training data set.

On the other hand, this study established a binary classifier that classifies only daily activities, fall and ADL; however, it is necessary to implement a robust performance model for posture detection that can classify each of the different kinds of daily activities. In particular, further study is needed on a model to classify between “fall” and “lie”, which are actions that can occur frequently in the daily lives of the elderly. Since these two behavior types are much more similar to each other than other behavior types, in order to train the differences between them, it is important to establish clear classification criteria for “lie” and “fall” and secure sufficient data to use for learning.

The performance of the algorithm used in this study can be improved by using deep learning techniques other than CNN or by using changed CNN in regard to structure. For example, as for long short-term memory (LSTM) of which recurrent neural network (RNN) architecture enhanced abled data processing and had the advantage of dealing with sequential data such as time series data. Therefore, it can be used to successfully and effectively classify ADL event and “Fall” event, which varies depending on time [51].

Along with other ambient sensor-based methods, an IR-UWB radar sensor was able to detect all features of the sensor arrangement environment [52]. The noise level of real living environment is higher than outdoor environment as there are many subjects that have an effect on the RF waveform such as furniture. Therefore, it is necessary to construct a more optimal model that can detect “Fall” and reflect the real living environment of seniors in future studies.

## Figures and Tables

**Figure 1 sensors-20-05948-f001:**
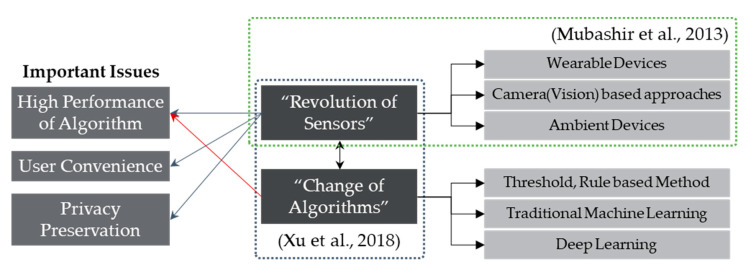
Classification of key issues and research topics related to fall detection.

**Figure 2 sensors-20-05948-f002:**
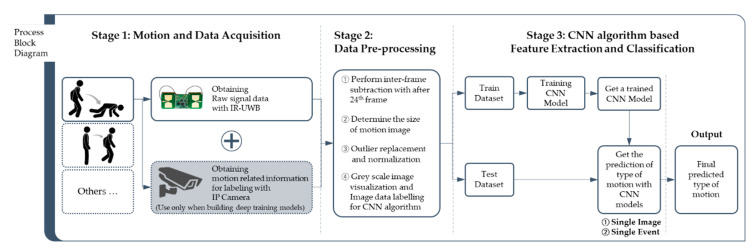
Proposed approach for fall detection using a convolution neural network.

**Figure 3 sensors-20-05948-f003:**
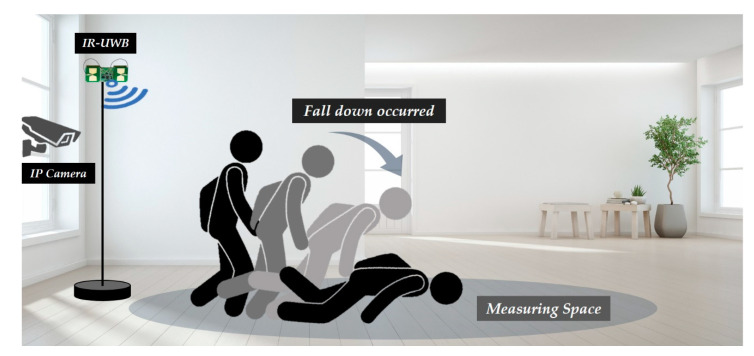
Conceptual drawing of object movement using IR-UWB radar sensor and IP camera.

**Figure 4 sensors-20-05948-f004:**
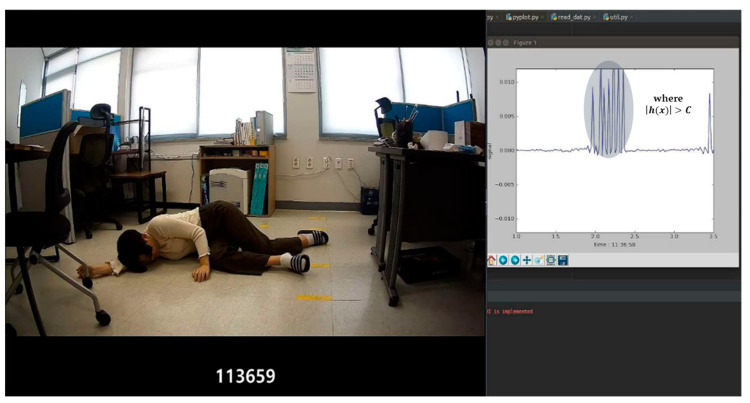
Visualized image of residual signal power value between current frame and the frame after 24th frame calculated according to Equations (1) and (2).

**Figure 5 sensors-20-05948-f005:**
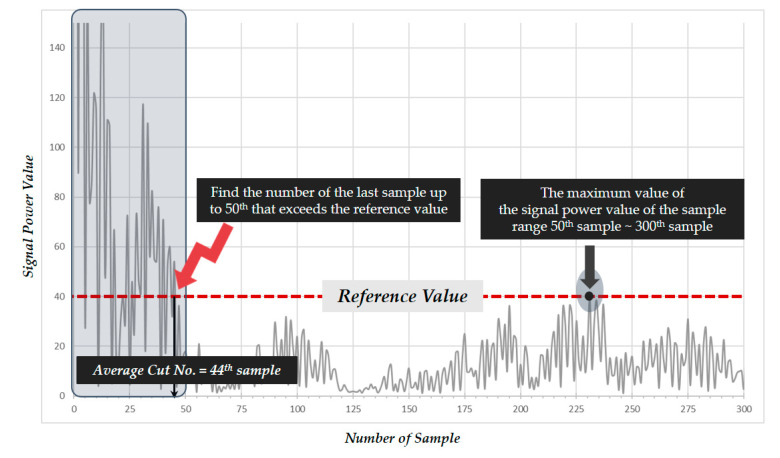
Labeling method considering motion window size and behavior type information.

**Figure 6 sensors-20-05948-f006:**
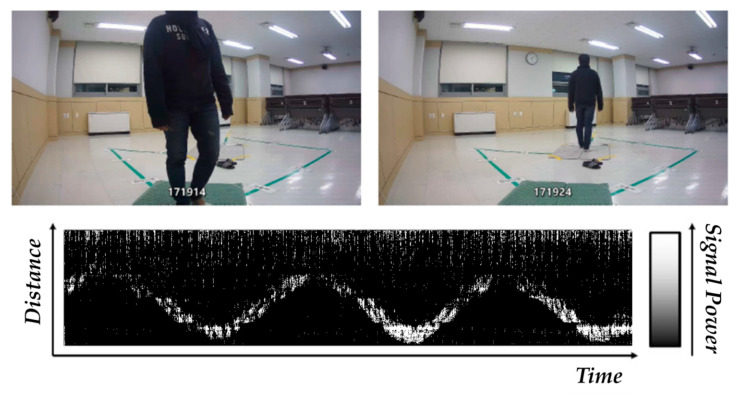
Example of image visualized by preprocessing of IR-UWB data.

**Figure 7 sensors-20-05948-f007:**
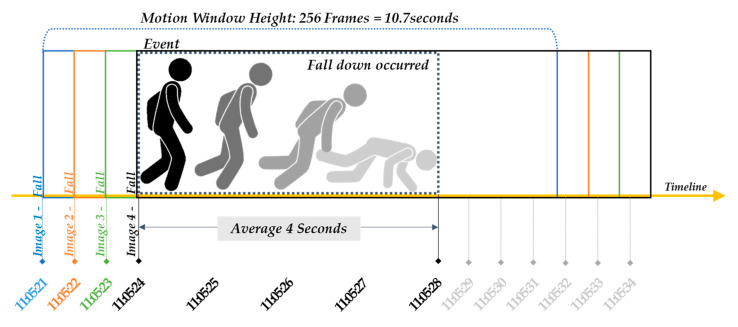
Labeling method considering motion window size and behavior type information.

**Figure 8 sensors-20-05948-f008:**
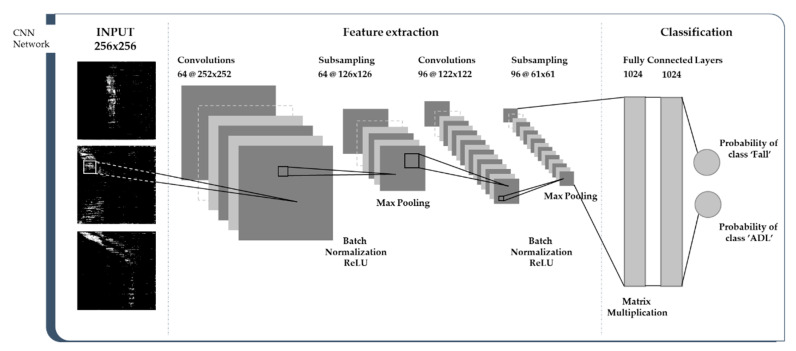
Architecture of convolution neural network (CNN) implemented for fall detection with classification of IR-UWB visualization image.

**Figure 9 sensors-20-05948-f009:**
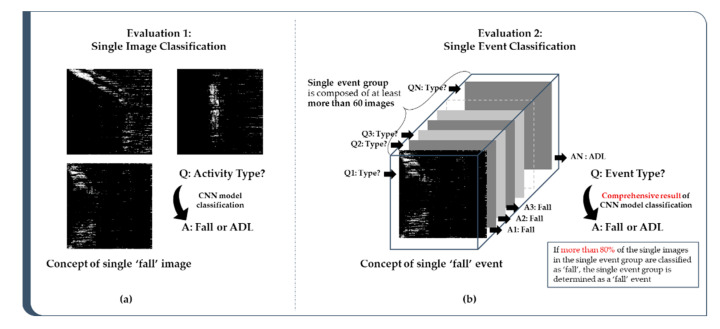
Concept difference between (**a**) single image classification and (**b**) single event classification.

**Figure 10 sensors-20-05948-f010:**
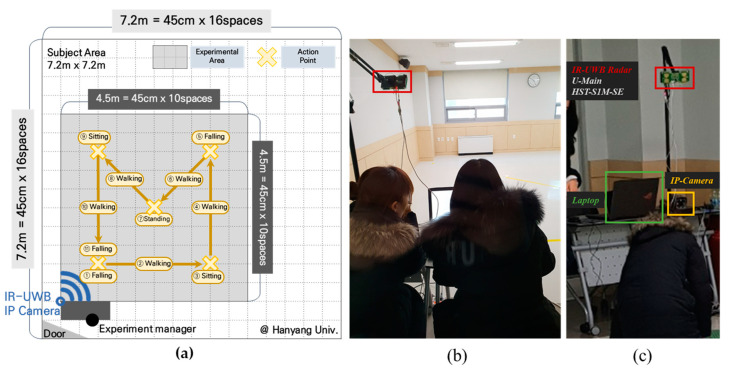
Informative diagram of data collection experimental environments: (**a**) sensor position, range of experimental area, action point, (**b**) the location of the test supervisor, (**c**) the location of the IR-UWB sensor, laptop, and IP camera.

**Figure 11 sensors-20-05948-f011:**
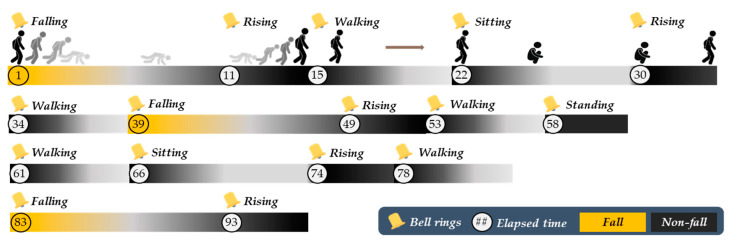
Scenario for building a database for creating a fall/activities of daily living (ADL) classification model.

**Figure 12 sensors-20-05948-f012:**
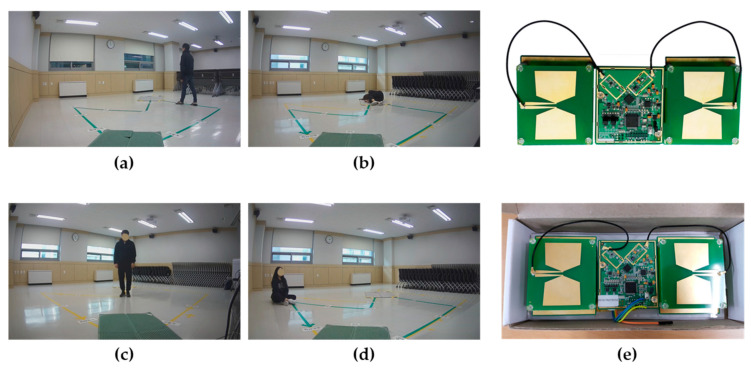
Pictures of different types of behavior performed in the experiment: (**a**) walking, (**b**) fall, (**c**) standing and (**d**) sitting and (**e**) U-MAIN HST-S1M-SE radar sensor.

**Figure 13 sensors-20-05948-f013:**
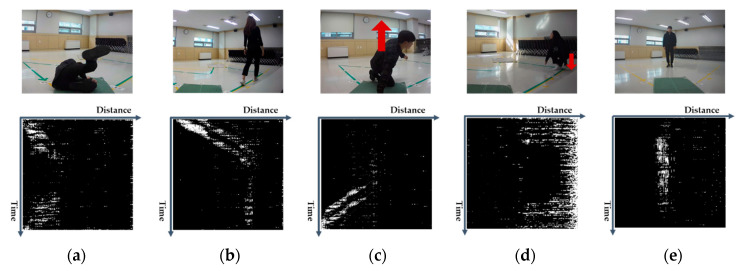
Results of motion image visualization: (**a**) fall, (**b**) walk, (**c**) rise, (**d**) sit, (**e**) stand.

**Figure 14 sensors-20-05948-f014:**
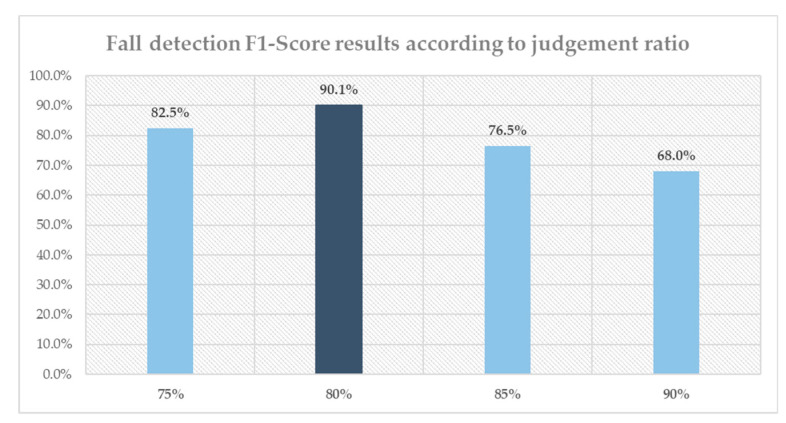
F1 score results of fall detection according to standard ratio.

**Table 1 sensors-20-05948-t001:** Advantages and disadvantages depending on sensor type.

Type of Sensor	Advantages	Disadvantages
Wearable Devices	- Privacy preservation - Cost-efficient - Ease of sensor installation and data collection - More accurate measurement of more activity indicators such as activity and heart rate	- Inconvenience due to wearing/attaching the sensor on a part of the body - Difficult to wear continuously due to the nature of the elderly, the primary users, and battery life is limited - Must be charged by a third party on site
Camera-based sensor	- Relatively high detection accuracy and robustness compared to other technologies - - Little disturbance to daily life (user convenience)	- Fatal weakness in terms of privacy - Degraded detection performance in dark environments - Detection is difficult outside the area where the sensor is installed
Ambient Sensor	- Privacy preservation - Little disturbance to daily life (user convenience)	- Easily exposed to noise - False alarms frequently occur - Deployment of the environmental sensors based systems is limited to indoor environments

**Table 2 sensors-20-05948-t002:** Class classification and detailed definition for each type of behavior.

Class	Behavior Type	Definition
ADL	Walk	Move at a regular pace by lifting and setting down each foot in turn, never having both feet off the ground at once
	Sit	The act of placing one’s body on the floor by weighting one’s hips in straight posture (one’s back is upright)
	Rise	The act of standing up straight after a “fall” or “sit”
	Stand	The act of staying straight with one’s legs stretched out against the ground for a while
Fall	Fall	Move downward, typically rapidly and freely without control, from higher (standing state) to a lower level

**Table 3 sensors-20-05948-t003:** Explanation of indicators for algorithm performance evaluation.

Indicator	Description	Calculation
True positive (TP)	A fall occurs, the system detects it	Count
True negative (TN)	A normal (ADL) movement is performed, the system does not declare a fall	Count
False negative (FN)	A fall occurs but the system does not detect it	Count
False positive (FP)	The system announces a fall, but it did not occur	Count
Accuracy (A)	Ratio of correctly predicted observations to total observations	$A=\frac{TP+TN}{Population}$
Recall (R)	Ratio of the positive observations correctly predicted to all observations in the actual class	$R=\frac{TP}{TP+FN}$
Precision (P)	Ratio of correctly predicted positive observations of the total predicted positive observations	$P=\frac{TP}{TP+FP}$
F1 score (F1)	Weighted average of precision and recall	$F1=\frac{2\left(R\times P\right)}{R+P}$

**Table 4 sensors-20-05948-t004:** Information of 6 participants.

Participant No.	Gender	Age	Height [cm]	Weight [kg]
1	Female	22	158	48
2	Female	22	161	60
3	Female	23	167	54
4	Male	21	165	63
5	Male	23	173	72
6	Male	24	175	69

**Table 5 sensors-20-05948-t005:** Product specification of HST-S1M-SE.

	Parameter	Value
Sensor Module	Detecting range	Up to 13 m
	Frequency range	3 ~ 4 GHz
	Bandwidth	0.45 ~ 1 GHz
	Output power	Typ. −25 dBM
	Distance resolution	2.03 cm
	Dimension	17.5.5 mm × 63.82 mm × 10 mm
Antenna Specification	Type	UWB directional antenna
	Gain	Avg. 7 dBi
	Antenna angle	56° (X-Z plane), 77.5° (Y-Z plane)
	Size	76 mm × 58.5 mm × 10 mm

**Table 6 sensors-20-05948-t006:** Hyper parameters values of CNN.

Hyper Parameter	Description
Convolution filter size	5 × 5
Stride	1
Epochs	100
Batch size	50
Dropout rate	10%

**Table 7 sensors-20-05948-t007:** Data configuration and class classification at the image level to create a CNN-based binary object behavior classification model.

	Type of Dataset	Class	Motion Type	Quantity	Ratio
Total Dataset (447,360)	Train dataset (339,288, 75.84%)	Fall (68,280, 20.12%)	Fall	68,280	20.12%
		ADL (271,008, 79.88%)	Walk	114,144	33.64%
			Stand	21,360	6.30%
			Sit	45,360	13.37%
			Rise	90,144	26.57%
	Test dataset (108,072, 24.16%)	Fall (22,176, 20.36%)	Fall	22,176	20.52%
		ADL (85,896, 79.48%)	Walk	34,992	32.38%
			Stand	7440	6.88%
			Sit	14,568	13.48%
			Rise	28,896	26.74%

**Table 8 sensors-20-05948-t008:** Event-level group creation results and configuration using the test dataset in Table 7.

	Class	Quantity	Ratio
Total events generated (1560)	Fall	282	18.08%
	ADL	1278	81.92%

**Table 9 sensors-20-05948-t009:** Results of algorithm performance evaluation at single image level: raw data.

		Predicted	
		Positive	Negative
Observed	Positive	(True Positive) 16436	(False Negative) 5740
	Negative	(False Positive) 5441	(True Negative) 80,455

**Table 10 sensors-20-05948-t010:** Results of algorithm performance evaluation at single image level: indicators.

Indicators	Value
Classification accuracy for training data	99.99%
Accuracy(A)	89.65%
Recall(R)	74.12%
Precision(P)	75.13%
F1 score (F1)	74.62%

**Table 11 sensors-20-05948-t011:** Results of algorithm performance evaluation at event level: raw data.

		Predicted	
		Positive	Negative
Observed	Positive	(True Positive) 260	(False Negative) 22
	Negative	(False Positive) 35	(True Negative) 1243

**Table 12 sensors-20-05948-t012:** Results of algorithm performance evaluation at event level: indicators.

Indicators	Value
Accuracy(A)	96.65%
Recall(R)	92.20%
Precision(P)	88.14%
F1 score (F1)	90.12%

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
