# Peer review of "IR-UWB Sensor Based Fall Detection Method Using CNN Algorithm"

_sensors, 2020, doi:10.3390/s20205948_

Round 1

Reviewer 1 Report

File attached

Author Response

  1. Section 1.3 though discusses about the proposed IR-UWB based solution and the contribution of this paper. Fall detection has been widely discussed. The fall detection using inertial sensors and machine learning algorithm have been widely tested in past few years. Authors have not clearly discussed the gaps or the limitations of the existing solutions. The contribution section does not clearly highlight the novelty of the work and how the proposed algorithm is better compared to other similar works.

1-1. Authors have not clearly discussed the gaps or the limitations of the existing solutions.

The types and limitations of existing solutions are mentioned according to the topics of sensors (section 1.1) and algorithms (section 1.2). Sensors are mentioned by three categories; wearable devices, camera-based sensors and ambient sensors. Also, advantages and disadvantages of the sensors are summarized in Table 1. Algorithms are mentioned which related to changes of algorithm’s development, such as algorithms based on threshold value which is the dominant before 2010, and the direction of development of machine learning based algorithms afterwards.

1-2. The contribution section does not clearly highlight the novelty of the work

We restate and emphasize the novelty of the work in the contribution section in section 1.4
(Line 230 ~ Line 229) & (Line 250 & Line 262)

1-3.  How the proposed algorithm is better compared to other similar works.

According to the comments, the content has been added as follows: Line 199 ~ 220 of section 1.3 introduces a study relevant to current “Fall” detection such as IR-UWB and described the pros and cons. Line 232 ~ 244 of section 1.4 refers to the benefits achieved by inputting two dimensional image incorporating the feature change of signal data effectively, which was transformed from an one dimensional time series of IR-UWB radar signal data.

  1. Section 2.2 describes the process of data collection with the help of volunteers younger than 25 years. It raises some practical concerns as the solution is proposed for elderly citizens and the data is collected with the help of young volunteers. Also, the details of data collections are insufficient.

2-1. It raises some practical concerns as the solution is proposed for elderly citizens and the data is collected with the help of young volunteers

The reasons why the test subjects were composed of young people in this study is related to safety accidents such as fractures that may occur when the actual elderly conduct the experiment. In fact, it is difficult and dangerous to collect enough fall event data from elderly, so the experiment was conducted after showing videos of the elderly falling to the test subjects several times in order to simulate as similar as possible.

As mentioned in Raul Igual’s review paper (2013), most of the fall studies are being conducted through simulations of young or mature people. According to Patricia Bet et al.(2019), of the last 20 years of papers surveyed in their review article, only 4.8% of studies dealt with the actual falls in older people over 60.

2-2. The details of data collections are insufficient.  

In section 2-2, we briefly introduced the overall contents related to data collection. Details such as the actual data collection environment, experimental scenario, and technical specifications of IR-UWB radar sensor used in the experiment were mentioned in Section 3; experimental setup. In reviewing your comment, we finally realized that the content was somewhat scattered, so it was not a structure that could be easily understood by the reader. In order to improve this problem and the reproducibility of the experiment, insufficient parts of data collection have been added and supplemented in Line 521 ~ Line 566. Complementary contents include the experiment supervisor, installation of actual sensor, camera and laptop (Figure 10 (b)-(c)), detailed information (gender, age, height, weight) of test subjects in Table 4, and the picture of sensor we actually used in the study (Figure 12e)

  1. The conceptual figure showing the object movement using IR-UWB and IP camera is a representative of a limited area. The effectiveness of the proposed solution is very limited.

In most previous studies, which detect falls using radio frequency based sensors such as IR-UWB, the maximum measuring distance of the sensor is mostly about 6~10m. The main cause can be said to be the limitation of the IR-UWB radar sensor’s measurement range. According to Quan, X et al. (2020) *, IR-UWB radar occupies a wide-band signal, it generally has to comply with stringent radio regulations. Therefore, IR-UWB radar is mainly used for indoor applications within a distance of 10m. Meanwhile, since older people always stay in a specific room and the size of the room is limited, the effective range can cover the range of activities of the elderly (**).

Type of Sensor

Distance from sensor

Title of Article

1

IR-UWB

Size of Single mattress

Fall detection in smart home environments using UWB sensors and unsupervised change detection.

2

IR-UWB

Fall @ 3.5m

Maximum 6m

Elderly-care motion sensor using UWB-IR

3

IR-UWB

Fall @ 1m and 4m

Maximum 5m

Analysis of an Indoor Biomedical Radar-Based System for Health Monitoring

4

IR-UWB

5m

A Radar-Based Smart Sensor for Unobtrusive Elderly Monitoring in Ambient Assisted Living Applications

5

IR-UWB

10m

Residual Network-Based Supervised Learning of Remotely Sensed Fall Incidents using Ultra-Wideband Radar

6

FMCW Radar

6m

Human Activity Classification With Radar: Optimization and Noise Robustness With Iterative Convolutional Neural Networks Followed With Random Forests.

[Reference]

* Quan, X., Choi, J. W., & Cho, S. H. (2020). A New Thresholding Method for IR-UWB Radar-Based Detection Applications. Sensors20(8), 2314.

** Wang, Y., Wu, K., & Ni, L. M. (2016). Wifall: Device-free fall detection by wireless networks. IEEE Transactions on Mobile Computing16(2), 581-594.

  1. Section-3 discusses experimental set-up. There are couple of concerns:
  2. Which IR-UWB sensor is used? There is no information about the sensor.

Information and detailed specification of the IR-UWB radar sensor which we used in the experiment were described in line 560 to 565 and Table 5. However, since there isn’t any photo of sensor actually used and installation situation for data measurement, it seems that the sensor-related contents were not conspicuous. So we added the photos in Figure 10(b)-(c) and Figure 12(e).

  1. CNN algorithm is not discussed except the figure-10 which shows CNN architecture.

It seems that except for the CNN architecture in Figure 10, deep-learning related contents are not sufficiently covered in the paper, so we reinforced the discussion about deep learning by newly creating Section 2.4.1; CNN architecture for training and classification and Section 2.4.2; Performance evaluation levels; single image and single event

  1. Images shown in section 4.1 are not very clear.

As per your opinion, we changed and added new image (Figure 13) with good (or clear) resolution in Section 4.1. In order to help understand the contents related to the generated images, photos for each situation were added and supplementary explanations were also added to Line 581 through Line 590. This is because the generated image alone makes it difficult for readers to intuitively capture the characteristics of the types of behaviors performed by the test subjects.

  1. The performance merit of the proposed CNN algorithm for classifying fall against non-fall is not compared with any other algorithms.

As mentioned in many review papers, most of the studies related to fall detection collected self-simulated data in a specific environment from volunteers with various characteristics that cannot be reiterated. Thus, it is hard to verify a given result or make a fair comparison with other studies. In other words, it is difficult to compare between each technology fairly unless data collected simultaneously by an identical subject in the identical environment for fall detection is guaranteed. However, Table 12 shows that the proposed model's numerical performance by using recent key indicators that evaluate the performance of the fall detection system. Compared to other major papers, it shows similar levels of performance in the accuracy, recall, and f1-score

  1. Igual, R.; Medrano, C.; Plaza, I. Challenges, issues and trends in fall detection systems. BioMedical Engineering Online, 2013, 12(1), 66.
  2. Ren, L.; Peng, Y., Research of fall detection and fall prevention technologies: A systematic review. IEEE Access, 2019, 7, 77702-77722.
  3. Islam, M. M.; Tayan, O.; Islam, M. R.; Islam, M. S.; Nooruddin, S.; Kabir, M. N.; Islam, M. R., Deep Learning Based Systems Developed for Fall Detection: A Review. IEEE Access, 2020, 8, 166117-166137

  1. References: except for 1 or 2 papers, most of the papers included in the list of references are old.

As you pointed out, we added various recent studies in the reference at Section 1.1 ~ Section 1.3 and so on. The reasons for inserting old references in the paper are as follows. As we reviewed our paper, we realized that there were some deficiencies related to literature review, and we have supplemented this overall.

In case of Load & Colvin(1991), Sposaro et al.(2009) and etc. are the first papers based on wearable sensor and smartphone, respectively, so we put those papers to mention first beginning and contents of the field of the study. Nouri et al. (2007), Igual et al. (2013), and Mubashir et al. (2013) were, as you know, the most representative papers in the field of fall detection research.

Reviewer 2 Report

This research aims to provide a new method to prevent falls, one of the most important health risk factors among elderly. The paper has an adequate scientific soundness, however, I think it could be improved:

  • I will appreciate a more detailed description of the experimental setup, including number of participants as well as their age, weight and height in order to improve its reproducibility. 
  • I miss a detailed study limits description.
  • I think that authors should discuss the advantages and disadvantages of their method against others (as IMU, accelerometers...)

Best regards

Author Response

  1. I will appreciate a more detailed description of the experimental setup, including number of participants as well as their age, weight and height in order to improve its reproducibility.

First of all, thanks for your comment. As you pointed out, insufficient parts of experimental setup have been added and supplemented in Line 520 ~ Line 577 to improve the reproducibility of the experiment. Complementary contents include the experiment supervisor, installation of actual sensor, camera and laptop (Figure 10 (b)-(c)), detailed information (gender, age, height, weight) of the test subjects in Table 4, and the picture of sensor we actually used in the study (Figure 12e). Moreover, it seems that except for the CNN architecture in Figure 8, deep-learning related contents are not sufficiently covered in the paper, so we also reinforced the discussion about deep learning by newly creating Section 2.4.1; CNN architecture for training and classification

  1. I miss a detailed study limits description.

We supplemented the detailed discussions on the limitations of the study in Section 5 (Line 660 ~ Line 674).

  1. I think that authors should discuss the advantages and disadvantages of their method against others.

In terms of sensor, Table 1 shows the advantages and disadvantages of sensor depending on the type. As for algorithm, section 1.2 shows the advantage of deep learning based algorithm compared to threshold value based method and traditional machine learning algorithm. Line 199 ~ 220 of section 1.3 introduces a study relevant to current “Fall” detection such as IR-UWB and described the pros and cons. Line 232 ~ 244 of section 1.4 refers to the benefits achieved by inputting two dimensional image incorporating the feature change of signal data effectively, which was transformed from one dimensional time series of IR-UWB radar signal data.

Round 2

Reviewer 1 Report

Kindly do proofreading and make some minor English corrections.

For example line 228-229, ……..which requires expertise in the time-frequency domain is required.

Reviewer 2 Report

I would like to acknowledge all the corrections that authors have made. In my opinion, the paper is improved and ready for its publication in Sensors